# Risk Factors for Post-Traumatic Stress Disorder after Childbirth: A Systematic Review

**DOI:** 10.3390/diagnostics12112598

**Published:** 2022-10-26

**Authors:** Ijlas El Founti Khsim, Mirella Martínez Rodríguez, Blanca Riquelme Gallego, Rafael A. Caparros-Gonzalez, Carmen Amezcua-Prieto

**Affiliations:** 1PhD Program in Clinical Medicine and Public Health, University of Granada, 18071 Granada, Spain; 2Department of Preventive Medicine and Public Health, Faculty of Medicine, University of Granada, 18016 Granada, Spain; 3Department of Nursing, University of Granada, 18071 Granada, Spain; 4Instituto de Investigación Biosanitaria (ibs.GRANADA), 18014 Granada, Spain; 5Consortium for Biomedical Research in Epidemiology and Public Health (CIBERESP), 28029 Madrid, Spain

**Keywords:** PTSD after birth, diagnosis, risk factors, pregnancy, perinatal mental health

## Abstract

Background: Post-traumatic stress disorder (PTSD) after birth has generated a growing interest in recent years. Although some risk factors associated with PTSD have been studied, information is still scarce to date on risk factors associated with PTSD. This systematic review aims to identify risk factors associated with the diagnosis of PTSD after childbirth. Methods: We searched on PubMed, Web of Science and SCOPUS databases, from inception to May 2022. Quality assessment of the articles was performed using the Newcastle-Ottawa Quality Assessment (“NOQAS”) scale. This systematic review was performed according to the PRISMA guidelines. Inclusion criteria were women with age ≥18 years; articles in English or Spanish; articles focused on physical, social, psychological, medical-obstetric, and environmental risk factors. Results: A total of *n* = 17,675 women were included among the studies in this systematic review. The main risk factors associated with PTSD after birth were obstetric interventions and obstetric violence such as emergency caesarean section or a non-compliant birth plan, a previous mental illness, having suffered from of a traumatic event or depression and/or anxiety, and having poor social support throughout pregnancy and/or during birth. Conclusions: Obstetric interventions, obstetric violence, experiencing a traumatic event or depression and/or anxiety, and a previous mental illness are factors associated with the diagnosis of PTSD after birth. Protective factors are multiparity, adherence to the mother’s birth plan and skin-to-skin contact.

## 1. Introduction

Post-Traumatic Stress Disorder (hereafter PTSD) is a disorder that occurs after suffering a stressful and extremely traumatic situation (e.g., physical and/or sexual abuse, a violent conflict, the death of a relative). During PTSD, the person experience a real danger to his or her life [1]. PTSD after birth affects between 3.1–6.3% of women after childbirth [2]. Pooled prevalence rates were 4.7% for PTSD and 12.3% for PTSS in mothers. Lower rates of 1.2% for PTSD and 1.3% for PTSS were found among fathers [3]. Besides, 30% of women often perceive their childbirth as a threat or trauma during the initial stage after childbirth [4].

PTSD after childbirth is considered the third most common psychiatric disorder among pregnant women, after depression and nicotine dependence [5]. Ten percent of women have symptoms of PTSD, although sometimes they do not comply with all the specific criteria for a formal diagnosis of PTSD after childbirth [4]. Despite being a highly prevalent disorder, according to Shields and Lyerly (2013), women are often excluded or severely underrepresented in clinical research [6]. Furthermore, it should be clear that the impact of PTSD after birth can have a negative impact on marriage and future reproductive choices [7]. This negative impact may be on children. All studies included in the systematic review by Van Sieleghem et al. [8], found a significant association between maternal PTSD and child behaviour/temperament, showing that children of mothers with PTSD had a more difficult temperament.

The symptomatology of PTSD after birth can increase during the first four to six weeks after childbirth. Five symptoms have been revised by psychologist Cheryl Beck, (1) persistent thoughts and memories of childbirth, through flashbacks and/or nightmares; (2) disconnection from their babies, which can appear immediately after the birth; (3) repetitive expression of what happened during childbirth; (4) feelings of anger towards professionals, family and self and (5) negative experience of motherhood [9]. By identifying and analysing factors that may lead to PTSD after childbirth, future interventions could be designed to reduce the risk of women experiencing childbirth as a traumatic process and/or to prevent the occurrence of PTSD after childbirth [10].

The aim of this systematic review was to identify risk factors associated with post-traumatic stress disorder after childbirth. The importance of conducting a systematic review on this topic lies in the scarcity of risk factors associated with PTSD after childbirth.

## 2. Materials and Methods

This systematic review was performed according to the PRISMA guidelines (Table A2) [11]. The PECO tool was applied to develop the research question. P—patients: women in their first, second or third trimester of pregnancy and postpartum; E—exposition: women exposed to PTSD risk factors (physical, social, psychological, medical-obstetric, and environmental risk factors); C—comparison: women non exposed to PTSD risk factors; O—outcome: post-traumatic stress disorder after birth; S—studies: cross-sectional, cohort, case-control, experimental and quasi-experimental studies.

Searches were carried out on Web of Science, Scopus and PubMed databases, from inception to May 2022. The following queries and MesH (Medical Subject Heading) terms were used: ((PTSD OR PTSD after childbirth) AND (mothers OR newborn)) OR ((PTSD OR PTSD after childbirth) AND (mothers OR newborn) OR ((PTSD OR PTSD after childbirth) AND (risk factors)) OR (PTSD OR PTSD after childbirth) AND (risk factors)) OR ((PTSD OR PTSD after childbirth) AND (traumatic birth) OR ((PTSD OR PTSD after childbirth) AND (traumatic birth)) OR ((PTSD OR PTSD after childbirth) AND (pregnancy and childbirth)) OR ((PTSD OR PTSD after childbirth) AND (pregnancy and childbirth)). In order to obtain a more precise search, additional terms were included in the search strategy: (PTSD [post-traumatic stress disorder], PTSD after birth, PTSD after delivery, postnatal post-traumatic stress disorder, traumatic birth, risk-factors, pregnancy, and childbirth). Studies were independently screened for eligibility and selected by two researchers (MMR y de IFK), as a result of a multi-step approach (removal of duplicates, reading title, abstract and assessment of the full text).

The article selection process was carried out by a first reviewer (M.M.R), who checked whether the study met the planned selection criteria, and discarded by title and/or abstract, and a second independent reviewer (I.F.K) Discrepancies between the two reviewers were resolved by discussion.

Similarly, data were extracted through independent work by two reviewers (I.F.K, M.M.R) and disagreements were cross-checked by a third reviewer (B.R.G).

### 2.1. Selection Criteria

Inclusion criteria were (a) women with an age ≥18 years who may suffer or have suffered from PTSD after childbirth; (b) articles in English or Spanish language; (c) articles focusing on physical, social, psychological, medical-obstetric, and environmental risk factors that may be related to the occurrence of PTSD after childbirth. Post-traumatic stress disorder is a persistent memory in which the traumatic experience is relieved. The person engages in avoidance behaviour and may show “hypervigilance”, to ensure that the traumatic event is not repeated. Symptoms persist for more than a month after the event and affect the person’s ability to function [12].

Exclusion criteria were (a) articles in languages other than English and Spanish; (b) articles whose results were not relevant; (c) papers whose full text was not available; (d) articles not available free of charge; (e) experimental animal studies were excluded.

### 2.2. Extraction, Synthesis of Data and Quality Assessment

Assessing the quality of such studies is essential for a proper understanding of non-randomised studies. As tabulation and description of the included reviews and articles addressing postpartum PTSD, authors of the study, year, country of study, research design, average age of participants, gestational age at delivery, risk factors associated with post-traumatic stress following childbirth and study quality Newcastle-Ottawa Quality Assessment Scale (NOQAS) [13]. This scale is used to review the quality of non-randomized studies (case-control and cohort studies). In the present systematic review, the NOQAS scale was used for quality review on selected cross-sectional studies [13]. The NOQAS is an ongoing collaboration between the Universities of Newcastle, Australia and Ottawa, Canada. It was developed to assess the quality of non-randomized studies with its design, content and ease of use directed to the task of incorporating the quality assessments in the interpretation of meta-analytic results. A ‘star system’ has been developed in which a study is judged on three broad perspectives: the selection of the study groups; the comparability of the groups; and the ascertainment of either the exposure or outcome of interest for case-control or cohort studies respectively. The goal of this project was to develop an instrument providing an easy and convenient tool for quality assessment of non-randomized studies to be used in a systematic review [13].

## 3. Results

Our search provided 8415 citations from inception to May 2022. We excluded 3561 duplicates. We rejected 3959 references because the titles or abstracts of the articles were unrelated to the objective to be investigated, leaving 57 full-text studies. Of these, 35 were excluded due to lack of relevant results, unavailability of full text or because they were in a language other than English or Spanish. Finally, 22 studies were selected for data extraction (Figure A1).

Obstetric interventions and obstetric violence were the main risk factors for developing PTSD after birth. Women with previous mental illness is also considered an important risk factor. The main social factor influencing the development of PTSD after childbirth is the low social support that women may have, both throughout the pregnancy process, and in the birth experience itself [14,15,16,17,18,19]. Table A1 provides a summary of the main characteristics of the selected studies, including the mean age of participants, the gestational stage of the study and the risk factors associated with post-traumatic stress following childbirth found in each study. Studies from different countries such as Canada, USA, England, Germany, Switzerland, Spain, Turkey, Iran, among others, were selected. Range age of participants was from 18 to 49 years. The sample size of the selected studies was from a minimum of *n* = 101 [7] to a maximum of *n* = 2990 [17]. One of the selected studies, a cross-sectional study, conducted in Iran, with *n* = 800 women, shows a positive association between fear of childbirth (OR = 3.47, CI 1.68–7.19) or lack of exercise during pregnancy (OR = 2.81, CI 1.40–5.63) with PTSD after birth [20]. A cohort study conducted in Switzerland, shows as the main risk factors for PTSD: having suffered maternal prepartum, intrapartum or postpartum complications (OR = 4.66 (1.54–14.06), anaesthetic complications (OR = 4.32; 95% CI = 1.04–17.87) and dissociative experiences during caesarean delivery (OR = 2.14 (1.08–4.25) [12]. Finally, in a prospective longitudinal study, conducted in Canada, higher anxiety sensitivity (OR = 1.75; 95% CI = 1.19–2.57), as well as history of sexual trauma (OR = 2.81; 95% CI = 1.07–7.37) were the determining factors of PTSD [21].

### 3.1. Risk Factors Associated with the Diagnosis of Post-Traumatic Stress Disorder after Childbirth

#### 3.1.1. Birth Experience

Experiencing a caesarean delivery, or an emergency delivery are the most influential risk factors (OR = 3.79; 95% CI = 2.43–5.92) [18] in triggering PTSD after childbirth [14,15,16,17,18,19,22]. In this context there is an association of obstetric interventions and obstetric violence with PTSD after birth [15,23]. Both third- and fourth-degree perineal tears have been associated with the development of PTSD after birth (OR = 2.77; 95% CI: 1.71–4.49) [22], as well as receiving uterine pressure or manual removal of the placenta (OR = 1.41; 95% CI = 1.03, 1.93) [17,18]. The feeling of fear of delivery is one of the major risk factors (OR = 3.47, 95%; CI = 1.68–7.19) [20], for developing PTSD after childbirth [7,16,20,21,24,25]. In addition, having a delivery plan that was not respected (OR = 2.85; 95%CI = 1.56–5.21) [23], or assessing birth as possibly traumatic (OR = 2.14; 95% CI = 1.08–4.25) [18], are important risk factors of PTSD after birth.

Childbirth complications, such as lower likelihood of co-homing, lack of skin-to-skin contact, and an absent of breastfeeding, have been associated with PTSD after birth [19]. These birth complications can be traced to problems in the newborn itself, such as newborn admission to Neonatal Intensive Care Units (NICU) [21,22,26,27], newborn admission to incubator [28], newborn admission to neonatal intermediate care, lower neonatal birth-weight [19], and infant experiencing actual or threatened injury [10]. Other factors related to PTSD are having a preterm birth (OR = 3.8; 95% CI = 0.27–53.16) [16], giving birth before arriving at hospital [24], primiparity (OR = 3.84; 95% CI = 0.99–2.11) [19], the Kristeller’s manoeuvre, lack of analgesic use during labour, or general anaesthesia [10,11,12,13,14,15,16,17].

#### 3.1.2. Psychological and Physical Risk Conditions

Women who experience depression and/or anxiety during pregnancy (OR = 1.75; 95% CI = 1.19–2.57,) [21], as well as those who have been diagnosed with a mental illness, are more likely to suffer from PTSD after childbirth [7,14,15,19,21,24,29]. Additionally, peritraumatic emotions and dissociations have been found to be strongly related to the occurrence of PTSD after birth, according to several studies [15,24,30]. Having poor sleep quality [28,31] and/or insomnia (OR = 3.00; 95% CI = 1.23–7.32) have been associated with PTSD [32].

#### 3.1.3. Social Risk Factors

Lack of maternal social support during pregnancy and/or during birth positively influence the development of PTSD after birth [10,16,21,25,28,32]. The experience of a lifetime traumatic event is also a social factor that could influence the occurrence of PTSD after birth [16,21,25,32]. Among these traumatic events, the risk is even higher when the trauma is due to any type of sexual violence (OR = 2.81; 95% CI = 1.07–7.37) [21,25,32]. An additional risk factor is the ethnic group (OR = 4.67; 95% CI = 1.72–14.74) [28] or having more specific sociodemographic characteristics, such as belonging to a non-European status or being single [15]. One study found that being a working woman could be a possible risk factor for developing this disease [29].

## 4. Discussion

This systematic review examined the main risk factors influencing the occurrence and development of PTSD after birth. PTSD after birth is a mental illness related to the birth experience, such as having had a caesarean section, emergency caesarean section and an obstetric violence experience [14,15,16,17,18,19,22].

In turn, the most common psychological factors that has been found to cause PTSD after birth are depression and/or anxiety throughout pregnancy [7,14,15,19,20,21,29]. Lack of social supported is the most important social risk factor for developing PTSD after birth, either during pregnancy and/or during birth [10,16,21,25,28,32]. Several studies have shown that depression and anxiety are important factors that can trigger mental illness [7,14,15,19,21,24,29]. Shaban et al. [29] established that women with moderate to severe anxiety suffered 3.5 times higher rates of PTSD after birth than women without anxiety. Similarly, anxiety is analysed as a significant predictor in the development of PTSD after birth [21].

Obstetric violence has been considered the main risk factors for the development of PTSD after birth. According to López et al., these different forms of obstetric violence can be a risk for both the mother and the baby, such as anaesthesia complications at delivery [15]. Women who suffered obstetric violence were more likely to suffer PTSD after birth [23]. In addition, emergency caesarean section and instrumental vaginal delivery are strongly associated with PTSD after birth [9]. Two recent systematic reviews have examined the relationship between mode of delivery and PTSD after birth. Cesarean delivery was more closely associated with PTSD than vaginal delivery (VD), emergency cesarean delivery (EmCB) more than elective cesarean delivery (ElCB), instrumental vaginal delivery (IVD) more than spontaneous vaginal delivery (SVD), and EmCB more than SVD. Emergency caesarean section and operative vaginal delivery have been found to be particularly associated with CB-PTS/D [33,34]. The systematic reviews from Ginter et al., 2022; Carter et al., 2022 and Heyne et al., 2022 were focused on the association between the type of delivery and its relationship with PTSD. Our review about PTSD risk factors, have considered social, psychological, obstetric, and medical factors [3,33,34].

Traumatic events seem to be a factor strongly associated with PTSD after birth [21,25]. PTSD develops after traumatic delivery and contains symptoms of reexperiencing, avoidance, negative cognitions and mood, and hyperarousal [35]. However, Shlomi et al. [7] recognized that previous traumatic events or sexual abuse, as well as the prevalence of previous psychiatric treatment were not significant in their occurrence.

In relation to demographic and social risk factors, Milosavljevic et al. [36] found no significant differences in demographic data between women who had or did not have PTSD after birth. Nevertheless, it seems that belonging to a non-European region or being single could influence the occurrence of PTSD after birth [15]. Verreault et al. [21] found that poor social support after childbirth was related to the development of PTSD after birth, while Zhang et al. [31] considered social support as a confounding factor.

It is important to highlight domestic/partner violence and its relevance in the occurrence of PTSD after birth. This factor remains understudied, but Oliveira et al. [37] have shown in their study that the frequency of at least one episode of both psychological and physical intimate partner violence during pregnancy was 71.95% and 21.2% respectively. Among women who developed PTSD after birth; 30.2% reported sexual violence in childhood and 92.5% and 45.0% suffered psychological and physical violence by their partners during pregnancy. According to recent studies, there is not enough scientific evidence to determine that cultural factors or preference for sons is related to PTSD, although it does seem to be related to higher levels of perinatal stress in the mother [38,39,40]. Emergency caesarean section is identified as a traumatic experience of childbirth and one of the main causes of postpartum post-traumatic stress disorder. There are high levels of PTSD (43.8%) in women after emergency caesarean section compared to elective caesarean section (23.2%) and other types of deliveries. Women with an emergency caesarean section are more vulnerable to developing PTSD because the emergency nature of the surgery [41]. Clara Heyne’s systematic review and meta-analysis focuses on measuring the mean prevalence of PTSD and PTSS. Among its criteria for the inclusion of studies, only those that reported a prevalence of any of the entities (PTSD or PTSS) were included, thus excluding those observational or experimental studies focused on the study of factors associated with PTSD. They used a study quality assessment tool that is unique to studies reporting prevalence. However, Newcastle-Ottawa Quality Assessment Scale (NOQAS) assesses the risk of bias in observational studies regardless of whether the study reported the prevalence of PTSD [3]. No direct association was found between the type of caesarean section and symptoms of postpartum depression. However, emergency caesarean section was found to be indirectly associated with symptoms of postpartum depression through symptoms of posttraumatic stress disorder. These findings suggest that unplanned surgical deliveries may be associated with the development of symptoms of posttraumatic stress disorder in mothers, which could subsequently lead to the development of symptoms of postpartum depression [42].

We have established some of the most statistically significant ORs to determine the variables most strongly associated with the development of PTSD, such as severe expected intensity of pain (OR = 4.65) [7], antepartum, intrapartum and postpartum complications (OR = 4.66) [15], complications with anaesthesia (OR = 4.32) [15], low social support (OR = 5.56) [10], fear of childbirth (OR = 3.47) [20] emergency caesarean section (OR = 3.58) [22], adult abuse (OR = 3.07) [32], two instances of MTE (OR = 4.48) [32], pregnancy-induced hypertension (OR = 5.04) [28], and presence of PPD symptoms (OR = 9.80) [28].

### 4.1. Systematic Review Strengths and Limitations

This systematic review tackles risk factors that may cause PTSD after birth. It has been carried out based on the PRISMA guidelines, establishing a PICO research question. A clear inclusion and exclusion selection criteria were defined, in order to be accurate. In turn, studies that were selected for analysis range from medium to high quality. Of these, 11 were of average quality [7,14,15,16,17,18,19,21,29,30,31,43] and other 11 of high quality [10,17,20,23,24,25,26,27,28,32,36].

Limitations of the review were the lack of information on the risk factors that can cause PTSD after birth. Studies included in the systematic review were mostly case-control (*n* = 11) and cohort studies (*n* = 2), including nine cross-sectional studies. Heterogeneity regarding the methodology of the included studies, and consequently from their findings, is somewhat evident. Heterogeneity is also true regarding different validity of the diagnoses [30,44] and outcomes variables reported [41,42]. Moreover, some studies did not provide a complete list of the exclusion criteria selected [14,22] so it seems they were not well conducted and hinders comparability between studies.

Another limitation of the systematic review was that even the protocol was prepared and applied for registration in the Open Science Framework (OSC), we still do not have a registration number.

### 4.2. Clinical Implications

Studies focused on the risk factors involved in PTSD after birth were from moderate to high quality. This systematic review emphasises the importance of knowing these risk factors for mental illness. Moreover, measures and actions can be taken throughout pregnancy to reduce factors involved in the development of PTSD after birth. The systematic review shows that obstetric violence or emergency caesarean section, are well known as a risk factors but other such as demographic situations, maternal morbidity, or insomnia, are lesser known. The inclusion of a short screening in a preconception consultation could greatly avoid PTSD after birth. In this screening variables that should be addressed are, among others, mother’s thoughts, women mental health (depression and/or anxiety), social support, fears about pregnancy and/or birth process.

Although there is not much information to date on what aspects are effective in preventing a traumatic birth experience, more direct support and care for women giving birth and their birth experience could be one of them [41].

### 4.3. Research Implications

Future research could include high-risk samples with the aim of understanding whether various predictive pathways can be applied, as well as considering different time points for collecting predictive data. Predictive data can identify risk before trauma occurs, this informing possible primary preventive measures. However, predictive data that is stored immediately after birth can be helpful, and more specifically, important for the creation of secondary preventive measures that can be administered after delivery [43].

On the other hand, it would be interesting for future research to perform a meta-analysis of the pooled effect sizes/odds ratios for the most important risk factors. This is a topic that could be addressed in future reviews when more information is available in this area.

## 5. Conclusions

Obstetric interventions, obstetric violence, depression and/or anxiety and previous mental illness are factors associated with the diagnosis of PTSD after birth. Protective factors are multiparity, adherence to the mother’s birth plan and skin-to-skin contact. Maternal physical and psychological factors are risk factors of PTSD after birth. Maternal social health, and social support, is even more important in the association with PTSD after birth occurrence. Further research and information will help to better understand and possibly prevent to reduce factors involved in its incidence.

## Data Availability

Not applicable.

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
