# Peer review of "Risk Factors for Post-Traumatic Stress Disorder after Childbirth: A Systematic Review"

_diagnostics, 2022, doi:10.3390/diagnostics12112598_

Round 1

Reviewer 1 Report

This review paper addresses an important topic in the field of women's mental health: the risk factors associated with the development of post-traumatic stress disorder (PTSD) following childbirth. I agree completely with the authors that this condition is often unrecognized and untreated, with potentially harmful consequences for both the mother and her child. 

The current review has several strengths: the authors have used a comprehensive search strategy, relying on several large and high-quality of databases (PubMed, Scopus, Web of Science), and have conducted their review in accordance with the PRISMA guidelines for systematic reviews.

The following aspects of the paper would benefit from correction or clarification to some extent:

1. The word "Supporting" in the title, abstract and text is somewhat misleading. From a careful reading of the paper, the authors' intent is to identify those factors associated with an elevated risk of PTSD in women following childbirth. A better phrase would be "risk factors for PTSD" or "factors associated with PTSD" or even "predictors of PTSD" if the evidence is strong enough.

2. A recent review by Heyne et al. (2022) has also examined the prevalence and risk factors associated with birth-related PTSD. It would be useful if the authors examine this review, compare the methods used, and clearly state the need for their own review or any novelties associated with it (for example, does it cover a wider range of literature, or provide a more comprehensive coverage of risk factors)? 

3. The review cited above could also help in providing an updated figure for the prevalence of PTSD and PTSD symptoms (PTSS) following childbirth.

4. The authors have mentioned the negative impact of PTSD after birth on marital relationships and reproductive choices. It would also be worth briefly discussing the potential negative impact of post-birth PTSD on children (reviewed in Van Sieleghem et al., 2022).

5. Two recent systematic reviews (Ginter et al., 2022; Carter et al., 2022) have examined the relationship between mode of birth and post-birth PTSD. As with the broader review by Heyne et al. (2022), the authors should consider the degree of overlap between these reviews and their own, discuss it in the course of the paper, and highlight the novel aspects / factors covered by their own review.

6. Was the review protocol pre-registered (e.g., PROSPERO, Web of Science)? This is not mandatory but should be mentioned (if done) or acknowledged as a limitation near the end of the Discussion (if not done).

7. Was the exclusion of papers in languages other than English / Spanish a significant limitation of this review? The authors have mentioned that 6 articles were excluded on this basis, which is a fairly large number given that the final review included only 22 papers.

8. I agree with the authors that the complete table of included studies and their quality is best presented as supplementary material. However, it is always useful, when presenting a systematic review, to at least provide a summary table of the key results / findings along with the text. This table could include the following headings: risk factor, studies supporting, studies with negative or opposing results, strength of association (pooled odds ratio, etc.) The authors could further divide this into birth-related and non-birth related factors (or proximal and distal risk factors) for ease of reading and conceptual clarity.

9. Under "social risk factors", it is surprising that this review does not mention the potential role of domestic violence / intimate partner violence as a risk factor for PTSD following birth. Though this subject is somewhat understudied due to the stigma / cultural factors / "code of silence" surrounding it in many countries, there is some empirical evidence that would merit inclusion. See, for example, Gankanda et al. (2021) or Oliveira et al. (2017); or the review by Geller and Stasko (2017). (The above were retrieved from PubMed; it is possible that other relevant studies could be found by searching the other databases.)

10. Was it possible to undertake a meta-analysis of pooled effect sizes / odds ratios for the most important risk factors? (If not, this should be mentioned as a subject for future reviews when more research in this area is available.)

11. Though these may not be available in the included studies, there are other factors that could be related to post-birth PTSD in specific cultures, such as gender preference (for a male child) or cultural practices following birth (which could be either protective or harmful). This aspect merits some coverage in the Discussion section given the global scope of this review.

12. From an examination of the supplementary table provided, it appears that all the included studies were published after 2012. Does this reflect a conscious choice by the reviewers (i.e., to include only studies published in the last 10 years?) If so, this should be mentioned in the review methodology. (If this was not by design, and there are no pre-2012 studies available, this point may be safely ignored.)

13. A certain degree of language editing is required to address minor errors in grammar and word choice.

Author Response

October 2022

Prof. Dr. Andreas Kjaer
Editor-in-Chief, Diagnostics

Department of Clinical Physiology, Nuclear Medicine & PET National University Hospital, Rigshospitalet, University of Copenhagen, Blegdamsvej 9, DK-2100 Copenhagen, Denmark

Manuscript ID:  Diagnostics-1958658

Manuscript title: Factors supporting the diagnosis of post-traumatic stress disorder after childbirth: a systematic review.

Dear Prof. Dr. Andreas Kjaer,

Thank you for your letter dated 07 October 2022. We were pleased to know that our manuscript was rated as potentially acceptable for publication in Diagnostics, Section: Pathology and Molecular Diagnostics, Special Issue: Diagnosis and Factors Associated with Perinatal Health, subject to adequate revision and response to the comments raised by the reviewers.

Sincerest thanks for your response and reviewers’ comments on our manuscript “Factors supporting the diagnosis of post-traumatic stress disorder after childbirth: a systematic review”. We hope that a revised version of the manuscript will still be considered by Diagnostics (Special Issue: Diagnosis and Factors Associated with Perinatal Health).

We have modified the paper in response to the insightful reviewer comments. I am therefore resubmitting the paper with the revisions that you suggested (using the “Track

Changes” function). 

Appended to this letter is our point-by-point response to the comments raised by the reviewers. As you may notice, we agreed with all the reviewers ‘comments and editor comments.

We would like to take this opportunity to express our sincere thanks to the reviewers who identified areas of our manuscript that needed corrections or modifications.

We would also like to thank you for allowing us to resubmit a revised version of our manuscript.

Yours sincerely,

Rafael A. Caparros-Gonzalez

Faculty of Health Sciences,

Department of Nursing

University of Granada, Granada, Spain.

Email: rcg477@ugr.es

Reviewers’ Comments

We would like to express our sincere thanks to the reviewers who identified areas of our manuscript that needed corrections or modifications. The reviewers´ comments will improve the quality of this manuscript.

Reviewer: 1

Comments to the Author
Manuscript ID:  Diagnostics-1958658

Manuscript title: Factors supporting the diagnosis of post-traumatic stress disorder after childbirth: a systematic review

Comments and Suggestions for Authors: This review paper addresses an important topic in the field of women's mental health: the risk factors associated with the development of post-traumatic stress disorder (PTSD) following childbirth. I agree completely with the authors that this condition is often unrecognized and untreated, with potentially harmful consequences for both the mother and her child.

Thank you for this comment. We are glad this reviewer found the topic of this paper interesting.

Comments and Suggestions for Authors: The current review has several strengths: the authors have used a comprehensive search strategy, relying on several large and high-quality of databases (PubMed, Scopus, Web of Science), and have conducted their review in accordance with the PRISMA guidelines for systematic reviews.

Thank you for this consideration.

Remark 1: The word "Supporting" in the title, abstract and text is somewhat misleading. From a careful reading of the paper, the authors' intent is to identify those factors associated with an elevated risk of PTSD in women following childbirth. A better phrase would be "risk factors for PTSD" or "factors associated with PTSD" or even "predictors of PTSD" if the evidence is strong enough.

We appreciate this comment. The title has been modified according to this reviewer suggestion and now reads: “Risk Factors for Post-Traumatic Stress Disorder After Childbirth: A Systematic Review”. The word "Supporting" has also been modified on page 1, line 3 of the abstract and in the text of the manuscript, on page 2, line 34.

Remark 2: A recent review by Heyne et al. (2022) has also examined the prevalence and risk factors associated with birth-related PTSD. It would be useful if the authors examine this review, compare the methods used, and clearly state the need for their own review or any novelties associated with it (for example, does it cover a wider range of literature, or provide a more comprehensive coverage of risk factors)?

This systematic review and meta-analysis aimed to determine mean estimates of prevalence rates for fulfilling all diagnostic criteria of posttraumatic stress disorder (PTSD) or at least showing significant levels of posttraumatic stress (PTSS) in relation to the traumatic event of childbirth. Inclusion criteria were specified for four different aspects. First, studies included in the synthesis had to report prevalence data from a quantitative, validated measure of PTSD/PTSS. Second, measurements had to be conducted within four weeks to 14 months postpartum. Third, study subjects had to be mothers who had experienced childbirth themselves and/or fathers/partners who attended birth. Childbirth was defined as live birth or stillbirth, the later referring to fetal death occurring after 20–24 weeks of pregnancy. The studies’ risk of bias was assessed using the JBI’s critical appraisal checklist .This checklist was developed specifically for evaluating studies reporting prevalence data. To be included in the quantitative analyses, a study had to have a minimum sample size of 20 for each subgroup. A study was considered as holding a low risk of bias if at least 60% of the JBI checklist’s items were rated as fulfilled. To improve the review, this information was added to the discussion section on page 5, line 238-244:“Clara Heyne's review and meta-analysis focuses on measuring the mean prevalence of PTSD and PTSS. Among its criteria for the inclusion of studies, only those that reported a prevalence of any of the entities (PTSD or PTSS) were included, thus excluding those observational or experimental studies focused on the study of factors associated with PTSD. They used a study quality assessment tool that is unique to studies reporting prevalence. However, Newcastle-Ottawa Scale (NOS) assesses the risk of bias in observational studies regardless of whether the study reported the prevalence of PTSD”.

Remark 3: The review cited above could also help in providing an updated figure for the prevalence of PTSD and PTSD symptoms (PTSS) following childbirth.

Thank you. To provide an updated figure for the prevalence of PTSD AND PTSD symptoms following childbirth, the mentioned review was included in the introduction on page 2, line 43-46: “Pooled prevalence rates were 4.7% for PTSD and 12.3% for PTSS in mothers. Lower rates of 1.2% for PTSD and 1.3% for PTSS were found among fathers”.

Remark 4: The authors have mentioned the negative impact of PTSD after birth on marital relationships and reproductive choices. It would also be worth briefly discussing the potential negative impact of post-birth PTSD on children (reviewed in Van Sieleghem et al., 2022).

Thank you for your suggestion. We have included this information on the negative impact on children in the introduction section, page 2, line 53-56:

"This negative impact may be on children. All studies included in the systematic review by Van Sieleghem et al., found a significant association between maternal PTSD and child behaviour/temperament, showing that children of mothers with PTSD had a more difficult temperament."

Remark 5: Two recent systematic reviews (Ginter et al., 2022; Carter et al., 2022) have examined the relationship between mode of birth and post-birth PTSD. As with the broader review by Heyne et al. (2022), the authors should consider the degree of overlap between these reviews and their own, discuss it in the course of the paper, and highlight the novel aspects / factors covered by their own review.

Thank you for the suggestion. We have included this information in the discussion section on page 5, line 203-213: “Two recent systematic reviews have examined the relationship between mode of delivery and PTSD after birth. Cesarean delivery was more closely associated with PTSD than vaginal delivery (VD), emergency cesarean delivery (EmCB) more than elective cesarean delivery (ElCB), instrumental vaginal delivery (IVD) more than spontaneous vaginal delivery (SVD), and EmCB more than SVD. Emergency caesarean section and operative vaginal delivery have been found to be particularly associated with CB-PTS/D.

The systematic reviews from Ginter et al., 2022; Carter et al., 2022 and Heyne et al., 2022 were focused in the association between the type of delivery and its relationship with PTSD. Our review about PTSD risk factors, have considered social, psychological, obstetric, and medical factors.”

Remark 6: Was the review protocol pre-registered (e.g., PROSPERO, Web of Science)? This is not mandatory but should be mentioned (if done) or acknowledged as a limitation near the end of the Discussion (if not done).

This comment has been included in the Discussion section, page 6, line 272-273: “The protocol was prepared, applied for registration in the Open Science Framework (OSF) and we still do not have a registration number.”

Remark 7: Was the exclusion of papers in languages other than English / Spanish a significant limitation of this review? The authors have mentioned that 6 articles were excluded on this basis, which is a fairly large number given that the final review included only 22 papers.

Thank you very much for your comment. The articles included in this review were in English and Spanish. This was an inclusion criterion in the systematic review. We have excluded other languages, but we do not consider that this is a significant limitation of the review.

Remark 8: I agree with the authors that the complete table of included studies and their quality is best presented as supplementary material. However, it is always useful, when presenting a systematic review, to at least provide a summary table of the key results / findings along with the text. This table could include the following headings: risk factor, studies supporting, studies with negative or opposing results, strength of association (pooled odds ratio, etc.) The authors could further divide this into birth-related and non-birth related factors (or proximal and distal risk factors) for ease of reading and conceptual clarity.

Thank you very much for your comment. We have considered to include Table A1, rather than as a supplementary table, in the manuscript, on page 9.

Remark 9: Under "social risk factors", it is surprising that this review does not mention the potential role of domestic violence / intimate partner violence as a risk factor for PTSD following birth. Though this subject is somewhat understudied due to the stigma / cultural factors / "code of silence" surrounding it in many countries, there is some empirical evidence that would merit inclusion. See, for example, Gankanda et al. (2021) or Oliveira et al. (2017); or the review by Geller and Stasko (2017). (The above were retrieved from PubMed; it is possible that other relevant studies could be found by searching the other databases.)

Thank you very much for contributing these articles that will improve our review. Although they do not meet the inclusion criteria, we have incorporated them into the discussion, page 5, line 226-231: “It is important to highlight domestic/partner violence and its relevance in the occurrence of PTSD after birth. This factor remains understudied, but Oliveira et al (2021) have shown in their study that the frequency of at least one episode of both psychological and physical intimate partner violence during pregnancy was 71.95% and 21.2% respectively. Among women who developed PTSD after birth; 30.2% reported sexual violence in childhood and 92.5% and 45.0% suffered psychological and physical violence by their partners during pregnancy”.

Remark 10: Was it possible to undertake a meta-analysis of pooled effect sizes / odds ratios for the most important risk factors? (If not, this should be mentioned as a subject for future reviews when more research in this area is available.)

The following paragraph was added to the discussion, specifically in the Research implications epigraph on page 7, line 296-298: “On the other hand, it would be interesting for future research to perform a meta-analysis of the pooled effect sizes/odds ratios for the most important risk factors. This is a topic that could be addressed in future reviews when more information is available in this area.”

Remark 11: Though these may not be available in the included studies, there are other factors that could be related to post-birth PTSD in specific cultures, such as gender preference (for a male child) or cultural practices following birth (which could be either protective or harmful). This aspect merits some coverage in the Discussion section given the global scope of this review.

Thanks for the suggestion, we researched this area more deeply and obtained the following information from different authors in the discussion section on page 5 and line 231-233: “According to recent studies, there is not enough scientific evidence to determine that cultural factors or preference for sons is related to PTSD, although it does seem to be related to higher levels of perinatal stress in the mother”.

Remark 12:  From an examination of the supplementary table provided, it appears that all the included studies were published after 2012. Does this reflect a conscious choice by the reviewers (i.e., to include only studies published in the last 10 years?) If so, this should be mentioned in the review methodology. (If this was not by design, and there are no pre-2012 studies available, this point may be safely ignored.)

We have included 22 studies, from inception to 2022. There were no temporal exclusion criteria, but by chance the studies selected were published between 2012 and 2022.

Remark 13: A certain degree of language editing is required to address minor errors in grammar and word choice.

Thank you, the document will be carefully reviewed for correction of grammatical errors.

The revision has been sent to a native speaker for more thorough proofreading.

Reviewer 2 Report

Considering that according to your comments in the Discussion “PTSD after birth is a mental illness related to the birth experience, such as having had a caesarean section, emergency caesarean section…” and related bibliography, caesarean section per se (including the scheduled one) could be a risk factor. This must be further commented and probably explained in your study.  

Author Response

October 2022

Prof. Dr. Andreas Kjaer
Editor-in-Chief, Diagnostics

Department of Clinical Physiology, Nuclear Medicine & PET National University Hospital, Rigshospitalet, University of Copenhagen, Blegdamsvej 9, DK-2100 Copenhagen, Denmark

Manuscript ID:  Diagnostics-1958658

Manuscript title: Factors supporting the diagnosis of post-traumatic stress disorder after childbirth: a systematic review.

Dear Prof. Dr. Andreas Kjaer,

Thank you for your letter dated 07 October 2022. We were pleased to know that our manuscript was rated as potentially acceptable for publication in Diagnostics, Section: Pathology and Molecular Diagnostics, Special Issue: Diagnosis and Factors Associated with Perinatal Health, subject to adequate revision and response to the comments raised by the reviewers.

Sincerest thanks for your response and reviewers’ comments on our manuscript “Factors supporting the diagnosis of post-traumatic stress disorder after childbirth: a systematic review”. We hope that a revised version of the manuscript will still be considered by Diagnostics (Special Issue: Diagnosis and Factors Associated with Perinatal Health).

We have modified the paper in response to the insightful reviewer comments. I am therefore resubmitting the paper with the revisions that you suggested (using the “Track

Changes” function). 

Appended to this letter is our point-by-point response to the comments raised by the reviewers. As you may notice, we agreed with all the reviewers ‘comments and editor comments.

We would like to take this opportunity to express our sincere thanks to the reviewers who identified areas of our manuscript that needed corrections or modifications.

We would also like to thank you for allowing us to resubmit a revised version of our manuscript.

Yours sincerely,

Rafael A. Caparros-Gonzalez

Faculty of Health Sciences,

Department of Nursing

University of Granada, Granada, Spain.

Email: rcg477@ugr.es

Reviewer: 2

Considering that according to your comments in the Discussion “PTSD after birth is a mental illness related to the birth experience, such as having had a caesarean section, emergency caesarean section…” and related bibliography, caesarean section per se (including the scheduled one) could be a risk factor. This must be further commented and probably explained in your study.

Thank you for your input, we have added information about caesarean section in the  "Discussion" section, page 5, line 187-190, line 202-213, line 234-250 we find this information about caesarean section and emergency ceasarean: “This systematic review examined the main risk factors influencing the occurrence and development of PTSD after birth. PTSD after birth is a mental illness related to the birth experience, such as having had a caesarean section, emergency caesarean section and an obstetric violence experience. In addition, emergency caesarean section and instrumental vaginal delivery are strongly associated with PTSD after birth. Two recent systematic reviews have examined the relationship  between mode of delivery and PTSD after birth. Cesarean delivery was more closely associated with PTSD than vaginal delivery (VD), emergency cesarean delivery (EmCB) more than elective cesarean delivery (ElCB), instrumental vaginal delivery (IVD) more than spontaneous vaginal delivery (SVD), and EmCB more than SVD. Emergency caesarean section and operative vaginal delivery have been found to be particularly associated with CB-PTS/D. Emergency caesarean section is identified as a traumatic experience of childbirth and one of the main causes of postpartum post-traumatic stress disorder. There are high levels of PTSD (43.8%) in women after emergency caesarean section compared to elective caesarean section (23.2%) and other types of deliveries. Women with an emergency caesarean section are more vulnerable to developing PTSD because the emergency nature of the surgery. No direct association was found between the type of caesarean section and symptoms of postpartum depression. However, emergency caesarean section was found to be indirectly associated with symptoms of postpartum depression through symptoms of posttraumatic stress disorder. These findings suggest that unplanned surgical deliveries may be associated with the development of symptoms of posttraumatic stress disorder in mothers, which could subsequently lead to the development of symptoms of postpartum depression.”

Reviewer 3 Report

Thank you for giving me the opportunity to review this manuscript,

I think it is necessary to revise the manuscript.

1) Please attach the PRISMA 2020 checklist and fill in the page numbers.

2) I think the PECO format is wrong. When the exposure is traumatic stress, and the outcome is the risk factor, their causal relationship is that the traumatic stress affects the risk factor later. Please make sure whether the PECO was clearly descrbed. Please revise it when necessary.

3) Please specify the methods used to decide whether a study met the inclusion criteria of the review, including how many reviewers screened each record and each report retrieved, whether they worked independently. How did the authors handle the situation when the two reviewers had different opinions.

4) Please specify the methods used to collect data from reports, including how many reviewers collected data from each report, whether they worked independently, any processes for obtaining or confirming data from study investigators.

5) Please provide registration information for the review, including register name and registration number, or state that the review was not registered.

6) Please explain where the review protocol can be accessed, or state that a protocol was not prepared.

7) Please describe the definition of PTSD clearly in the method section. In the eligibility criteria section, you showed the word of PTSD without any explanation of what is PTSD. Was it a diagnostic criteia of specific mannual or just some symptoms?

7) I think the many risk factors of post-traumatic stress after childbirth were not always clinically meaningful, because the odds ratio were very small (around one odds ratio). What kind of risk factors were clinically meaningful? How many odds ratio means the risk factors clinically meaningful?

I think it is necessary to revise the manuscript before publication.

Author Response

October 2022

Prof. Dr. Andreas Kjaer
Editor-in-Chief, Diagnostics

Department of Clinical Physiology, Nuclear Medicine & PET National University Hospital, Rigshospitalet, University of Copenhagen, Blegdamsvej 9, DK-2100 Copenhagen, Denmark

Manuscript ID:  Diagnostics-1958658

Manuscript title: Factors supporting the diagnosis of post-traumatic stress disorder after childbirth: a systematic review.

Dear Prof. Dr. Andreas Kjaer,

Thank you for your letter dated 07 October 2022. We were pleased to know that our manuscript was rated as potentially acceptable for publication in Diagnostics, Section: Pathology and Molecular Diagnostics, Special Issue: Diagnosis and Factors Associated with Perinatal Health, subject to adequate revision and response to the comments raised by the reviewers.

Sincerest thanks for your response and reviewers’ comments on our manuscript “Factors supporting the diagnosis of post-traumatic stress disorder after childbirth: a systematic review”. We hope that a revised version of the manuscript will still be considered by Diagnostics (Special Issue: Diagnosis and Factors Associated with Perinatal Health).

We have modified the paper in response to the insightful reviewer comments. I am therefore resubmitting the paper with the revisions that you suggested (using the “Track

Changes” function). 

Appended to this letter is our point-by-point response to the comments raised by the reviewers. As you may notice, we agreed with all the reviewers ‘comments and editor comments.

We would like to take this opportunity to express our sincere thanks to the reviewers who identified areas of our manuscript that needed corrections or modifications.

We would also like to thank you for allowing us to resubmit a revised version of our manuscript.

Yours sincerely,

Rafael A. Caparros-Gonzalez

Faculty of Health Sciences,

Department of Nursing

University of Granada, Granada, Spain.

Email: rcg477@ugr.es

Reviewer: 3

Comments and Suggestions for Authors

Thank you for giving me the opportunity to review this manuscript,

I think it is necessary to revise the manuscript.

Remark 1: Please attach the PRISMA 2020 checklist and fill in the page numbers.

Thanks for the suggestion, we have included the PRISMA table as Table A2, in the Appendix section, page 8.

Remark 2: I think the PECO format is wrong. When the exposure is traumatic stress, and the outcome is the risk factor, their causal relationship is that the traumatic stress affects the risk factor later. Please make sure whether the PECO was clearly described. Please revise it when necessary.

Thank you for your clarification, the suggested change has been made in the section "Material and methods", page 2, line 70-75: “The PECO tool was applied to develop the research question. P - patients: women in their first, second or third trimester of pregnancy and postpartum; E - exposition: women exposed to PTSD risk factors (physical, social, psychological, medical-obstetric, and environmental risk factors); C-comparison: women non exposed to PTSD risk factors; O - outcome: post-traumatic stress disorder after birth ; S - studies: cross-sectional, cohort, case-control, experimental and quasi-experimental studies.”

Remark 3: Please specify the methods used to decide whether a study met the inclusion criteria of the review, including how many reviewers screened each record and each report retrieved, whether they worked independently. How did the authors handle the situation when the two reviewers had different opinions.

The selection method for the inclusion of the different studies has been added in the     "material and methods" section, page 3, line 89-92: The article selection process was carried out by a first reviewer (I.F.K) who checked whether the study met the planned selection criteria, and discarded by title and/or abstract, and a second independent reviewer.(M.M.R). Discrepancies between the two reviewers were resolved by discussion.”

Remark 4: Please specify the methods used to collect data from reports, including how many reviewers collected data from each report, whether they worked independently, any processes for obtaining or confirming data from study investigators.

Thank you for your comment, the changes have been added in "Materials and methods", page 3, line 93-94:

"Similarly, data were extracted by work done independently by two reviewers (I.F.K, M.M.R) and disagreements were cross-checked by a third reviewer (B.R.G)."

Remark 5: Please provide registration information for the review, including register name and registration number, or state that the review was not registered.

It has been specified in the Discussion section, on page 6, line 272-273: “ The protocol was prepared, applied for registration in the Open Science Framework (OSF) and we still do not have a registration number.”

Remark 6: Please explain where the review protocol can be accessed, or state that a protocol was not prepared.

This comment has been included in the limitations section of the Discussion, page 6, line 272-273: “ The protocol was prepared, applied for registration in the Open Science Framework  (OSF) and we still do not have a registration number.”

Remark 7: Please describe the definition of PTSD clearly in the method section. In the eligibility criteria section, you showed the word of PTSD without any explanation of what is PTSD. Was it a diagnostic criteia of specific mannual or just some symptoms?

Thank you for your comment, in the introduction we have defined the meaning of post-traumatic stress disorder and specified that from now on the acronym would be used for its understanding (PTSD). However, a definition of PTSD has been added again by the MSD manual in the section on inclusion criteria, under "Methods and materials", page 3, line 99-102: Post-traumatic stress disorder is a persistent memory in which the traumatic experience is relieved. The person engages in avoidance behaviour and may show "hypervigilance", to ensure that the traumatic event is not repeated. Symptoms persist for more than a month after the event and affect the person's ability to function.”

Remark 8: I think the many risk factors of post-traumatic stress after childbirth were not always clinically meaningful, because the odds ratio were very small (around one odds ratio). What kind of risk factors were clinically meaningful? How many odds ratio means the risk factors clinically meaningful?

Thank you for your comment. We have added the following information in the discussion section, page 6, line 251-257: "We have established some of the most statistically significant ORs for the variables most strongly associated with the development of PTSD, such as severe intensity of expected pain (OR= 4.65) [7], complications before, during and after delivery (OR= 4. 66) [15], complications with anaesthesia (OR= 4. 32) [15], poor social support (OR= 5.56) [10], fear of childbirth (OR= 3.47) [20], emergency caesarean section (OR= 3. 58) [22], adult abuse (OR= 3.07) [32], two cases of MSD (OR= 4.48) [32], pregnancy-induced hypertension (OR= 5.04) [28], and presence of PPD symptoms (OR= 9.80) [28]". 

Round 2

Reviewer 3 Report

I think this manuscript would be suitable for publication in this journal.